# First Report of the Production of Mycotoxins and Other Secondary Metabolites by *Macrophomina phaseolina* (Tassi) Goid. Isolates from Soybeans (*Glycine max* L.) Symptomatic with Charcoal Rot Disease

**DOI:** 10.3390/jof6040332

**Published:** 2020-12-03

**Authors:** Vivek H. Khambhati, Hamed K. Abbas, Michael Sulyok, Maria Tomaso-Peterson, W. Thomas Shier

**Affiliations:** 1Department of Biochemistry, Molecular Biology, Entomology, and Plant Pathology, Mississippi State University, Mississippi State, MS 39762, USA; vhk4@msstate.edu (V.H.K.); MariaT@pss.msstate.edu (M.T.-P.); 2Biological Control of Pests Research Unit, US Department of Agriculture, Agricultural Research Service, Stoneville, MS 38776, USA; 3Institute of Bioanalytics and Agro-Metabolomics, Department of Agrobiotechnology, IFA-Tulln, University of Natural Resources and Life Sciences, Vienna (BOKU), Konrad-Lorenz-Str. 20, Tulln 3430, Austria; michael.sulyok@boku.ac.at; 4Department of Medicinal Chemistry, College of Pharmacy, University of Minnesota, Minneapolis, MN 55455, USA; shier001@umn.edu

**Keywords:** moniliformin, kojic acid, mellein, orsellinic acid, cyclo[L-proline-L-tyrosine], cordycepin, phaseolinone, botryodiplodin, natural products, phytotoxins, secondary metabolites, fungi

## Abstract

*Macrophomina phaseolina* (Tassi) Goid., the causal agent of charcoal rot disease of soybean, is capable of causing disease in more than 500 other commercially important plants. This fungus produces several secondary metabolites in culture, including (-)-botryodiplodin, phaseolinone and mellein. Given that independent fungal isolates may differ in mycotoxin and secondary metabolite production, we examined a collection of 89 independent *M. phaseolina* isolates from soybean plants with charcoal rot disease using LC-MS/MS analysis of culture filtrates. In addition to (-)-botryodiplodin and mellein, four previously unreported metabolites were observed in >19% of cultures, including kojic acid (84.3% of cultures at 0.57–79.9 µg/L), moniliformin (61.8% of cultures at 0.011–12.9 µg/L), orsellinic acid (49.4% of cultures at 5.71–1960 µg/L) and cyclo[L-proline-L-tyrosine] (19.1% of cultures at 0.012–0.082 µg/L). In addition, nine previously unreported metabolites were observed at a substantially lower frequency (<5% of cultures), including cordycepin, emodin, endocrocin, citrinin, gliocladic acid, infectopyron, methylorsellinic acid, monocerin and N-benzoyl-L-phenylalanine. Further studies are needed to investigate the possible effects of these mycotoxins and metabolites on pathogenesis by *M. phaseolina* and on food and feed safety, if any of them contaminate the seeds of infected soybean plants.

## 1. Introduction

Soybean (*Glycine max* L.) is an economically valuable crop due to its high protein and oil content and wide variety of uses in food, agricultural pharmaceutical and industrial applications [1]. The United States produces 32.5% of the world’s soybeans (120 million tons/year valued at $31.2 billion), making it the second most valuable US crop [2]. In the southern US, high temperatures and dry conditions, along with poor management, can lead to drought-stressed plants that are susceptible to infection by opportunistic soil- and seed-borne pathogens, such as *Fusarium virguliforme* O’Donnell & T. Aoki, *Cercospora sojina* Hara, *Phomopsis longicolla* T.W. Hobbs and *Macrophomina phaseolina* (Tassi) Goid., resulting in significant economic impact due to yield loss and reduced seed quality [3,4,5,6,7,8,9,10]. *M. phaseolina* infects over 500 species of commercially important plants, causing various dry-weather wilts and rots, including charcoal rot disease in soybean [6,7,8,9,10]. In many years, charcoal rot disease is the major cause of soybean crop yield loss and seed quality deterioration in the southern US states of Arkansas, Louisiana and Mississippi [11,12,13], although soybean cyst nematode is the major cause of crop damage in most parts of the country. It is not understood what factors specific to the region are responsible for the relatively greater pathogenicity of *M. phaseolina* there [14].

One plausible explanation for the greater frequency of charcoal rot disease in soybeans in this region is that *M. phaseolina* isolates endemic to the region produce unusual levels or diversity of mycotoxins. Therefore, as an approach to exploring the possible role of mycotoxin production in the increased pathogenicity of *M. phaseolina*, the present study was undertaken to investigate the diversity of known mycotoxins and other secondary metabolites produced in culture by isolates from regional soybean plants symptomatic for charcoal rot using liquid chromatography tandem mass spectrometry (LC-MS/MS) with culture medium filtrates. *M. phaseolina* exhibits extensive genetic diversity as assessed by DNA markers and differences in pathogenicity [15,16,17,18]. The genetic diversity of *M. phaseolina* may extend to mycotoxins, given that (-)-botryodiplodin (1) and not phaseolinone (3) was found [19,20,21] in soybeans with charcoal rot (Figure 1), whereas phaseolinone (3) (Figure 1) but not (-)-botryodiplodin was isolated from the culture medium of an *M. phaseolina* endophyte from mung bean [22]. Similarly, mellein (Figure 1) has been observed in some but not all isolates of *M. phaseolina* from soybean plants symptomatic of charcoal rot in Mississippi, U.S.A. [23].

## 2. Materials and Methods

### 2.1. M. phaseolina Culture Collection and Sources

For this study, 89 isolates of *M. phaseolina* were obtained from soybean plants exhibiting symptoms of charcoal rot disease in various fields in Mississippi and the surrounding areas using the method described by Mengistu et al. [9], or they were provided by other investigators in the same areas [20,21] (see Figure 2). Cultures were maintained on potato dextrose agar (PDA) until use.

### 2.2. Preparation of Cell-Free Culture Extracts

*M. phaseolina* isolates were cultured in Czapek-Dox broth (CDB) prepared by dissolving 2.0 g NaNO_3_, 1.0 g K_2_HPO_4_, 0.5 g KCl, 0.5 g MgSO_4_, 0.01 g FeSO_4_ and 30 g sucrose in 1 L water. Aliquots (200 mL) of CDB in 500 mL Erlenmeyer flasks were sterilized in an autoclave for 15 min at 121 °C then cooled. Each flask was inoculated with five 5 mm mycelial plugs from a 7-day-old PDA culture of each *M. phaseolina* isolate. Inoculated flasks fitted with cotton plugs were incubated in an Innova 40 Benchtop Incubator Shaker (New Brunswick Scientific Co., Inc., Edison, NY, USA) for 10 days at 28 °C and 110 rpm. After incubation, the liquid cultures were filtered roughly through cheesecloth to remove large biomass not able to pass through a fine filter. Cell-free filtrates (CFF) of the *M. phaseolina* isolates were prepared by vacuum filtration in Nalgene Rapid-Flow sterile disposable filter units with 0.45 µm pore-size CN membrane filters (Thermo Fisher Scientific, Rochester, NY, USA) and freeze-dried. Dried CFF powders were stored at −20 °C until shipped for chemical analysis studies.

### 2.3. Chemicals and Reagents for LC/MS Analyses

Reagents for LC-MS analysis included HiPerSolv Chromanorm HPLC gradient grade acetonitrile from VWR Chemicals (Vienna, Austria), LC-MS Chromasolv grade methanol from Honeywell (Seelze, Germany) and LC-MS grade ammonium acetate and glacial acetic acid from Sigma-Aldrich (Vienna, Austria). Reverse osmosis water was purified through a Purelab Ultra system (ELGA LabWater, Celle, Germany) for analytical use. Mellein was purchased from Cayman Chemical (Ann Arbor, MI, USA); botryodiplodin was prepared as described in Abbas et al. [21]. The detailed sources of the remaining analytical standards are listed in the supplementary information (ESM 2) of Sulyok et al. [26].

### 2.4. LC-MS/MS Analysis

Samples were analyzed by LC-MS/MS as described by Sulyok et al. [26] using an extension of the method to include botryodiplodin and mellein (Table 1). Briefly, a 1290 series UHPLC system (Agilent Technologies, Waldbronn, Germany) was connected with a QTrap 5500 MS/MS system (Sciex, Foster City, CA, USA) containing a TurboV electrospray ionization source (Sciex, Foster City, CA, USA). A Phenomenex (Torrance, CA, USA) Gemini C18 column (150 × 4.6 mm i.d. and 5 µm particle size) was utilized for chromatographic separation along with a Phenomenex C18 security guard cartridge (4 × 3 mm i.d.). An injection volume of 5 µL with a flow rate of 1 mL/min in binary gradient mode was used for elution. Both mobile phases were methanol/water/acetic acid with Eluent A 10:89:1 (*v*/*v*/*v*) and Eluent B 97:2:1 (*v*/*v*/*v*) and contained 5 mM ammonium acetate. Initially, 100% A was held for 2 min; afterwards, B was increased linearly to 50% over 3 min. B was then linearly increased to 100% over 9 min and B was held at 100% for 4 min. Finally, the column was re-equilibrated back to 100% A within 2.5 min. Scheduled multiple reaction monitoring (sMRM) was used for analyte detection. Quantification of the analytes was based on external calibration using a serial dilution of a multi-component stock solution. The accuracy of the method was verified on a continuous basis by participation in ring trials organized by BIPEA (Genneviliers, France) with a current success rate (z-scores between −2 and +2) of approximately 95% for over 1400 results submitted [26]. For confirmation of a positive identification, the ion ratio had to agree with the corresponding values of the standards within 30% as stated in the SANTE/12089/2016 document [27], whereas for the retention time, a stricter criterion of ± 0.03 min was applied in this study.

### 2.5. Statitical Analysis

The frequencies of occurrence of mycotoxins and other secondary metabolites in the study were analyzed for significant differences by pairwise applications of Fisher’s exact test to rank-ordered frequencies; *p* < 0.05 was considered significant.

## 3. Results

### Identification and Frequency of Secondary Metabolites

Cell-free culture filtrates from a total of 89 *M. phaseolina* isolates from soybean plants symptomatic for charcoal rot were analyzed by LC-MS/MS. In addition to the known *M. phaseolina* metabolites, (-)-botryodiplodin (1) and mellein (2) (Figure 1), the analysis detected a total of 13 secondary metabolites (Figure 3 and Figure 4) that have not previously been reported to be produced in culture by *M. phaseolina* [19,20,21,22,23]. Several metabolites (Figure 3) were observed at a frequency in the range of 19% to 84%, as well as the previously reported metabolites (-)-botryodiplodin (1) (82.0%) and mellein (2) (28.1%). Kojic acid (4), which was produced by 84.3% of isolates in the concentration range of 0.57–79.9 µg/L, was the most frequently observed metabolite (Table 2). Other previously unreported metabolites were moniliformin (5) (61.8% of isolates), orsellinic acid (6) (49.4% of isolates) and cyclo(L-proline-L-tyrosine) (7) (i.e., maculosin) (19.1% of isolates).

The LC-MS/MS analysis also detected the production in the culture media of nine previously unreported metabolites by *M. phaseolina* at relatively low frequency (less than 5%) among the 89 isolates in the collection (Table 3). The low frequency metabolites were cordycepin (8), emodin (9), endocrocin (10), citrinin (11), gliocladic acid (12), infectopyron (13), methylorsellinic acid (14), monocerin (15) and N-benzoyl-phenylalanine (16) (Figure 4).

## 4. Discussion

The large number of secondary metabolites produced by *M. phaseolina* and the differences in the metabolite profiles of various isolates presumably reflect the genetic diversity of the fungus [15,16,17,18]. The diversity of toxins may play a role in the reported differences in pathogenicity of *M. phaseolina* isolates [17,18] if any of the toxins are used in the root infection mechanism or to weaken plant defenses [10,19,20,21,22,23,28].

Among the *M. phaseolina* metabolites identified for the first time in this study, kojic acid, moniliformin, orsellinic acid and mellein are the most notable. Kojic acid is a metabolite originally isolated from *Aspergillus oryzae* (Ahlb.) Cohn that has found commercial use as a skin-whitening agent in cosmetic products [29,30]. Kojic acid has been reported to have mutagenic properties at high concentrations in several animals, but the amount found in food is generally safe for human consumption [31,32]. Kojic acid acts by inhibiting the enzyme tyrosinase [33], which is important for pigment formation and defense mechanisms in plants, animals and insects. Moniliformin usually occurs as a sodium or potassium salt. It was initially purified from corn contaminated with *Fusarium moniliforme* J. Sheld., later classified as *F. proliferatum* (Matsush.) Nirenberg ex Gerlach & Nirenberg [34]. It has subsequently been shown to be produced by a variety of *Fusarium* spp. that contaminate cereal grains worldwide, including *F. moniliforme*; *F. sporotrichioides*, Sherbakoff, 1915; *F. avenaceum*, (Fr.) Sacc., 1886; *F. culmorum*, (W. G. Sm.) Sacc. 1895 and *F. oxysporum* Schltdl., 1824 [35]. Moniliformin production by *Penicillium melanoconidium* (Frisvad) Frisvad & Samson, 2004, which also contaminates cereal grains, has also been reported [36]. Moniliformin has been extensively studied due to its high acute toxicity in both plants and animals, particularly cardiotoxicity in poultry [37,38]. Moniliformin is believed to act by inhibiting mitochondrial respiration by inhibiting pyruvate and α-ketoglutarate oxidation [39]. Orsellinic acid has been isolated from various species of fungi [39,40], but few studies have explored the biological activities of the underivatized molecule; published reports suggest that the compound has little to no toxic activity [40,41]. Mellein, also known as ochracin, is a well-known mycotoxin initially extracted from *A. melleus* Yukawa in 1933 and subsequently shown to be produced by numerous organisms, including fungi, plants, bacteria, and insects [42,43]. In 2018, Khambhati et al. [22] analyzed several *M. phaseolina* cultures by gas chromatography–mass spectrometry and discovered that the fungus synthesized mellein as well as other unidentified substances. In 2020, Salvatore et al. [44] reported the production of mellein by an *M. phaseolina* isolate from *Eucalyptus globulus* Esser, Lora L. Mellein is reported to have numerous biological activities, including phytotoxicity, cytotoxicity, antifungal activity and others [36]. Cyclopeptides with antifungal activity, such as cyclo(L-proline-L-tyrosine), are produced by a wide range of bacterial and fungal species [45].

The discovery that a series of mycotoxins and other secondary metabolites are produced in culture by pathogenic isolates of *M. phaseolina* raises several issues requiring further study. Additional studies will be needed to determine if any of the major metabolites identified in cultures of *M. phaseolina* impact the pathogenesis of the fungus in soybean and if any accumulate in the seeds at sufficient levels to affect yield or food safety. Of particular concern are food safety issues that would arise if any of these secondary metabolites accumulate in seeds or other plant parts that are consumed as food or animal feed. In the case of (-)-botryodiplodin, Abbas et al. [19] found detectable levels of the toxin only in the roots of soybeans with charcoal rot disease, which is not a food safety concern, given that soybean roots are not used for food or feed. However, more extensive studies are needed to determine if, and under what conditions, each of the mycotoxins and other secondary metabolites identified in cultures of *M. phaseolina* accumulate at toxic levels in seeds or other edible plant parts for soybeans and other food and feed plants susceptible to *M. phaseolina* infection. The newly identified *M. phaseolina* metabolites may also provide tools to better understand the pathology of charcoal rot disease in soybeans and other commercially important species. Future studies may include expanded studies on the incidence of the newly discovered secondary metabolites and phytotoxic evaluations in experimental systems, such as hydroponic culture, monitoring toxicity symptoms including chlorosis, necrosis, wilting, stunting and/or death [21].

## Figures and Tables

**Figure 1 jof-06-00332-f001:**
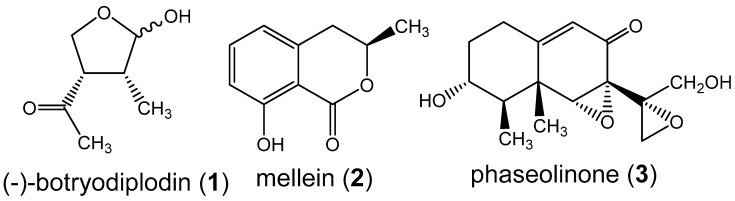
Structures of some previously reported toxins from *Macrophomina phaseolina* cultures, (-)-botryodiplodin (**1**), mellein (**2**) and phaseolinone (**3**).

**Figure 2 jof-06-00332-f002:**
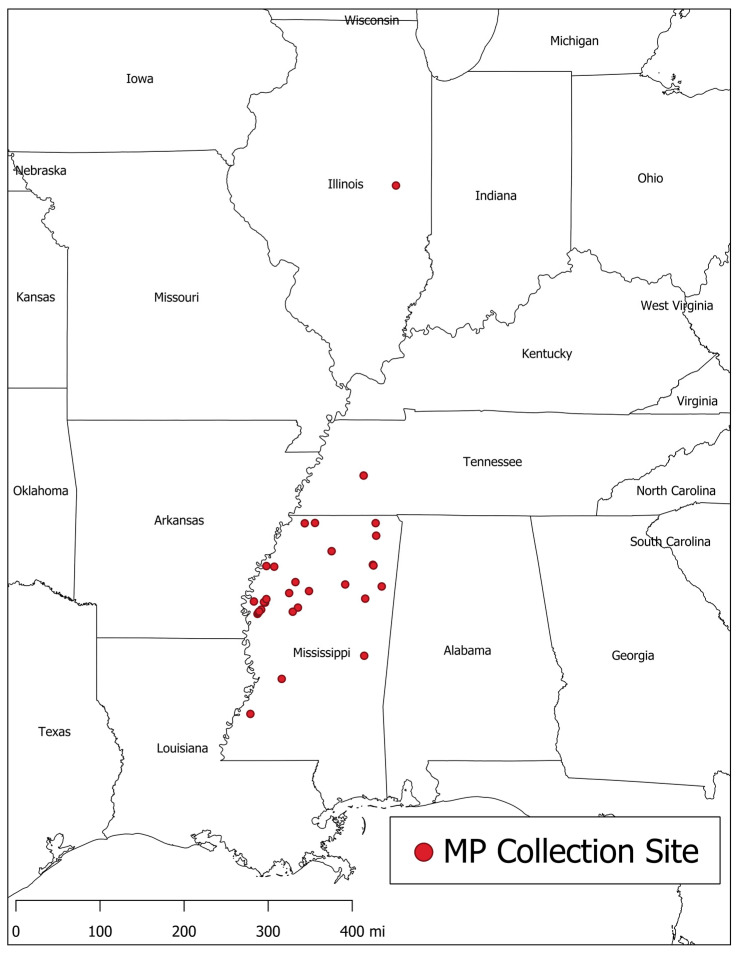
Map of *M. phaseolina* collection sites in the United States. Each point represents a site where one or more isolates of *M. phaseolina* were collected. There were 31 collection sites in total. The map was generated using QGIS software [24] and cartographic boundary files were provided by the US Census Bureau [25].

**Figure 3 jof-06-00332-f003:**
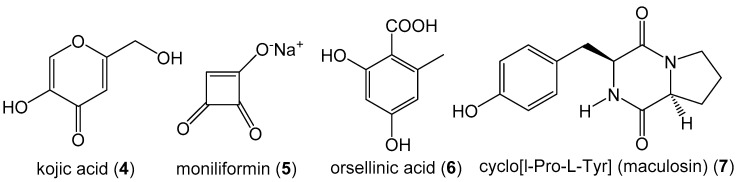
Structures of mycotoxins and secondary metabolites produced in culture media by *M. phaseolina* in more than 19% of the 89 *M. phaseolina* isolates from soybean plants symptomatic for charcoal rot.

**Figure 4 jof-06-00332-f004:**
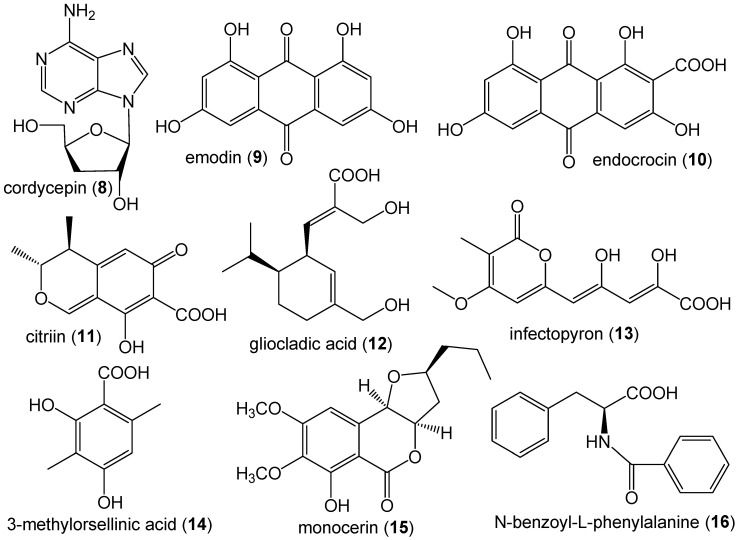
Structures of mycotoxins and secondary metabolites produced in the culture media by *M. phaseolina* in less than 5% of the 89 *M. phaseolina* isolates from soybean plants symptomatic for charcoal rot.

**Table 1 jof-06-00332-t001:** LC-MS/MS parameters for botryodiplodin and mellein.

Compound	tR ^a^ (min)	*m*/*z* Q1	DP ^b^	*m*/*z* (Q3)	CE ^c^ (V)	CXP ^d^ (V)
Botryodiplodin	4.7	127.1	61	83.0/85.0	17/17	6/6
Mellein	9.1	179.1	96	105.1/115.1	32/8	40/8

^a^ retention time; ^b^ declustering potential; ^c^ collision energy; ^d^ cell exit potential.

**Table 2 jof-06-00332-t002:** Mycotoxins and other secondary metabolites occurring in the culture media of a relatively high percentage (>19%) of the 89 *M. phaseolina* isolates from soybean plants symptomatic for charcoal rot included in the study.

*M. phaseolina* Metabolite	% of Isolates Producing the Metabolite ^1^	Range of Metabolite Concentrations Produced (µg/L)	Mean ± SEM Concentration (µg/L) of Producers
Kojic acid (4)	84.3% a	0.57–79.9	13.3 ± 18.6
(-)-Botryodiplodin (1)	82.0% a	15.9–2380	865 ± 651
Moniliformin (5)	61.8% b	0.011–12.9	1.03 ± 1.69
Orsellinic acid (6)	49.4% c	5.71–1960	347 ± 412
Mellein (2)	28.1% e	1.66–74.9	12.5 ± 10.9
Cyclo[L-Pro-L-Tyr] (7)	19.1% e	0.012–0.082	0.038 ± 0.017

^1^ Values followed by the same letter in Table 2 and Table 3 do not have significantly different frequencies (*p* < 0.05, Fisher’s exact test).

**Table 3 jof-06-00332-t003:** Mycotoxins and other secondary metabolites occurring in the culture media of relatively few (<5%) of the 89 *M. phaseolina* isolates from soybean plants symptomatic for charcoal rot included in the study.

*M. phaseolina* Metabolite	Number of Isolates Producing the Metabolite ^1^	Range of Metabolite Concentrations Produced (ng/L)
Cordycepin (8)	4 f	42.2–90.4
Emodin (9)	2 f	0.34–67.3
Endocrocin (10)	2 f	15.3–1180
Citrinin (11)	1 f	10.5
Gliocladic acid (12)	1 f	35.7
Infectopyron (13)	1 f	1960
Methylorsellinic acid (14)	1 f	840
Monocerin (15)	1 f	5.1
N-Benzoyl-L-phenylalanine (16)	1 f	0.68

^1^ Values followed by the same letter in Table 2 and Table 3 do not have significantly different frequencies (*p* < 0.05, Fisher’s exact test).

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
