# Peer review of "First Report of the Production of Mycotoxins and Other Secondary Metabolites by Macrophomina phaseolina (Tassi) Goid. Isolates from Soybeans (Glycine max L.) Symptomatic with Charcoal Rot Disease"

_jof, 2020, doi:10.3390/jof6040332_

Round 1

Reviewer 1 Report

Macrophomina phaseolinaisolated, a fungus, causes charcoal rot disease affecting soybean crop yield and seed quality. The authors have performed non-targeted metabolomic analysis of a collection of 89 independent M. phaseolinaisolated from soybean with charcoal rot disease in various fields in Mississippi and surrounding areas. Out of six metabolites that at relatively high frequency (>19% of cultures), four of them were reported at the first time.

The work is technically competent, and represents a useful report of the metabolite profiles of this fungus causing disease in soybean and other crops. However, without following up biological relevance, it contributes to the identification of the metabolites, rather than to the pathogenesis of this fungus or to the soybean. Is there any relation between the existent or concentration of these six major metabolites with the yield or seed quality of soybean?

Other points.

- It is not clear how the metabolite concentrations produced and the concentration of producers (Table 1, 2) were calculated.

- In the methods section, it would be great to have a map showing the locations for sampling. It will be helpful for following studies and investigators.

Reviewer 2 Report

Dear Authors,

I have an honour to make a revision of the manuscript Brief communication entiled: “ First Report of the Production of Mycotoxins and Other Secondary Metabolites by Macrophomina phaseolina (Tassi) Goid. Isolates from Soybeans (Glycine max L.) Symptomatic with Charcoal Rot Disease” submitted to Journal of Fungi MDPI.

The authors presented data that show several interesting findings on the examined a collection of 89 independent M. phaseolina isolates from soybean plants with charcoal rot disease using LC-MS/MS analysis of culture filtrates. Although the manuscript seems to contain a few of valuable information on the emerging soil-borne and seed-borne pathogen Macrophomina phaseolina, but it should be improved by paying attention to the following:

  • Please change the key words to avoid phrase staying in the title like Macrophomina phaseolina or mycotoxins of the manuscript.

  • Please explain in the introduction a bit more the fact that phaseolina infects over 500 species of commercially important plants causing various dry weather wilts and rots – please adjust information not only for U.S. Soybean plant are economically important for sure, please underlined why authors decided to choose that kind of host.

  • Author’s did not explain, what is the difference between M. phaseolina pathogen isolates and phaseolina exist as endophytic isolates. Is endophytic isolates excluded infection with other pathogenic isolates?

  • Please underline more the aim of the studies.

  • Authors declare: “four previously unreported metabolites were observed at relatively high frequency (>19% of cultures)” – the question is, does 19 % could be explain as high percentage? – please explain.
  • Although authors presented nine previously unreported metabolites were observed at relatively low frequency (<5% of cultures), as they presented, including cordycepin, emodin, endocrocin, citrinin, gliocladic acid, infectopyron, methylorsellinic acid, monocerin, and N-benzoyl-L-phenylalanine, the real problem is that Authors did not revealed any statistical analysis of presented quantitative results.

  • The discussion style is poor, please rewrite it. Please rethink the results and it’s applicable used and summarize own results in the context of recent publications. The reader has an impression that the results were organized in manuscript in hurry and without deep analysis.

Sincerely,

Author Response

Reviewer #2:

  1. As requested by Reviewer #2, the Keywords mycotoxin and Macrophomina phaseolina have been removed and replaced with cyclo[L-Pro-L-Tyr] and cordycepin to avoid repeating the title.

  1. Reviewer #2 requested that we explain in the introduction in more detail why we focus on soybeans when M phaseolina infects over 500 species of commercially important plants. In response to this request we have added the following sentence at p. 2, l. 51, original manuscript: “It is not understood what factors specific to the region are responsible for the relatively greater pathogenicity of M. phaseolina there.”

  1. Reviewer #2 sought clarification concerning whether or not endophytic isolates were excluded from the study. To address this issue, the sentence “It was not determined if they had infected plants from the soil reservoir or were endophytes.” was added at p. 2, l. 70, original MS. 

  1. Reviewer #2 requested clarification about the aim of the studies. To address this issue, a new paragraph was created, and the Aims sentence was moved from the last sentence to the first sentence of the new paragraph. Also, additional explanation was added to the sentence in the form of the phrase “as an approach to exploring the possible role of mycotoxin production in increased pathogenicity of M. phaseolina”.  The enhanced sentence is now located at p. 2, l. 51, original MS and reads as follows: 

Therefore, as an approach to exploring the possible role of mycotoxin production in increased pathogenicity of M. phaseolina, the present study was undertaken to investigate the diversity of known mycotoxins and other secondary metabolites produced in culture by isolates from soybean plants symptomatic for charcoal rot using liquid chromatography tandem mass spectrometry (LC-MS/MS) with culture medium filtrates.

  1. Reviewer #2 felt that describing occurrence in “>19% of cultures” as being “at relatively high frequency” needed explanation. This concern has been addressed in the revised MS by deleting “high frequency” and replacing it with “were observed in >19% of cultures,” on p.1, ll. 26-27, original manuscript and adding “at a substantially lower frequency” on p. 1, l. 30, original manuscript. 

  1. Reviewer #2 requested that statistical analysis of the data be presented. To address this issue, statistical analysis of the frequency of occurrence of the various toxins has been added to Tables 1 and 2 along with a footnote as follows:  “Values followed by the same letter in Tables 1 and 2 are not significantly different frequencies (P<0.05, Fisher’s exact test).” The modifications are incorporated at p. 4, ll. 128-132 (Table 1) and p. 4, ll. 140-142 (Table 2), original manuscript.  

  1. Reviewer #2 felt the Discussion needed improvement. To address this issue, additional discussion of mechanisms of action of the commonest new mycotoxins for which the mechanism is known has been added with references. 

(1) The following text was added at p. 5, l. 153, original MS:

Kojic acid acts by inhibiting the enzyme tyrosinase [32], which is important for pigment formation and defense mechanisms in plants, animals and insects.

(2) The following text was added at p. 5, l. 161, original MS:

Moniliformin is believed to act by inhibiting mitochondrial respiration by inhibiting pyruvate and α-ketoglutarate oxidations [38].

The following references have been added: 

  1. Chang, T.-S. An updated review of tyrosinase inhibitors. Int J Mol Sci. 2009, 10, 2440-2475, doi: 10.3390/ijms10062440.
  2. Thiel, P.G. A molecular mechanism for the toxic action of moniliformin, a mycotoxin produced by Fusarium moniliforme. Biochem. Pharmacol., 1978, 27, 483-486.

The necessary changes in reference numbers have been made where needed elsewhere in the MS.  This includes changes in reference numbers to correct an error found in numbering original reference numbers 39 (Sulyok et al.) and 40 (SANTE), which now appear as references 25 and 26. 

Round 2

Reviewer 1 Report

All previous suggestions are satisfied in this version.

Author Response

There were no comments in the second round of revisions from the reviewer.

Reviewer 2 Report

Dear Authors,

I have an honour to make a revision of the manuscript Brief communication entiled: “ First Report of the Production of Mycotoxins and Other Secondary Metabolites by Macrophomina phaseolina (Tassi) Goid. Isolates from Soybeans (Glycine max L.) Symptomatic with Charcoal Rot Disease” submitted to Journal of Fungi MDPI.

The authors presented data that show several interesting findings. But I am sorry to say that in “improved version of manuscript” in the most points ignored my questions and concerns about the manuscript. Response for reviewer questions is not deleting sentence in any means. When reviewer ask questions he wants clarification in the text of manuscript or in response. In most of part new version of article authors completely ignored my comments simply by avoiding them. I am sorry to say but I cannot accept manuscript and I changed my recommendation for it.

I points that was not explain by authors:

  1. “Please explain in the introduction a bit more the fact that phaseolina infects over 500 species of commercially important plants causing various dry weather wilts and rots – please adjust information not only for U.S. Soybean plant are economically important for sure, please underlined why authors decided to choose that kind of host. “

Added sentence by authors is completely in appropriate for my request. I wanted information about worldwide economic problems with this pathogen not only in US. Is important for Journal of Fungi MDPI to present worldwide context of research not local as for US. Is well known Macrophomina phaseolina  is one the most dangerous pathogen but not for soybean is over 50 papers about this problematic. This is the reason why I want explanation why only soybean was choose for research if Macrophomina  is not major pathogen of this plant. Authors did not add any information which I requested

  1. “Author’s did not explain, what is the difference between M. phaseolina pathogen isolates and phaseolina exist as endophyticIs endophytic isolates excluded infection with other pathogenic isolates?”

I am sorry to say but expiation used by authors is strange. If is not known if this fungus is endphyte or pathogen. Why authors starts research the first place about this interaction? If  this fungus a endophyte then it cannot be pathogen. If it is pathogen then rather not endpohyte. If this matter is not known how authors could performed research about their isolates if they cannot distinguished if this isolates are pathogens.

  1. “Please underline more the aim of the studies.”

I am again sorry to say but aim of the study is not underline and clarified at all . Authors only moved sentence in text of manuscript. Aim of the study is still fuzzy and It must be rewrite.

  1. “Authors declare: “four previously unreported metabolites were observed at relatively high frequency (>19% of cultures)” – the question is, does 19 % could be explain as high percentage? – please explain.”

Authors ignored my comment they deleted only used terms. Why 19% is high procentege?

  1. “Although authors presented nine previously unreported metabolites were observed at relatively low frequency (<5% of cultures), as they presented, including cordycepin, emodin, endocrocin, citrinin, gliocladic acid, infectopyron, methylorsellinic acid, monocerin, and N-benzoyl-L-phenylalanine, the real problem is that Authors did not revealed any statistical analysis of presented quantitative results.”

Authors add minimal information about statistic with adding letters in two tables. But in material and methods authors did not information about statistic methodology used.

New issue

The numeration of table are not according journal rules. Tables must be cited as the are pointed out in the text so it should be Table 1,2,3 etc.

Now authors starts numeration of tables from Table 3 and next is 1 and 2 it must be corrected.

Author Response

Dear Reviewer #2,

We have made additional revisions to our Brief Report entitled "First Report of the Production of Mycotoxins and Other Secondary Metabolites by Macrophomina phaseolina (Tassi) Goid. Isolates from Soybeans (Glycine max L.) Symptomatic with Charcoal Rot Disease" by Vivek Hemant Khambhati, myself, Michael Sulyok, Maria Tomaso-Peterson, Wayne Thomas Shier.  The revisions are described below.  Attached is the manuscript revised to include the reviewer suggestions.  We hope that you will find it satisfactory in its present form. 

The following changes have been made. 

Reviewer #2:

  1. Reviewer #2 requested that we explain in the introduction in more detail why we focus on soybeans when M phaseolina infects over 500 species of commercially important plants.

The real reason why we are studying pathogenic isolates of M phaseolina from soybeans mostly in Mississippi, USA, is that the research has been funded by soybean growers from Mississippi.  We have indicated this on p. 7, ll. 219-220, revised manuscript, as follows:  

“Funding: This research was funded by the Mississippi Soybean Promotion Board and supported by the Mississippi Agricultural and Forestry Experiment Station . . . “

The reason why the soybean growers from Mississippi provide their own funds for research on M. phaseolina is that in many years charcoal rot disease is the Number 1 cause of yield and quality loss for soybeans from the three US states of Arkansas, Louisiana, and Mississippi. 

We felt that we had made that fact clear with the following sentence in the original manuscript: 

“In many years charcoal rot disease is the major cause of soybean crop yield loss and seed quality deterioration in the southern U.S. states of Arkansas, Louisiana, and Mississippi [11-13].”

In the second revision we have expanded the sentence by the suggestion of the reviewer as follows: 

“In many years charcoal rot disease is the major cause of soybean crop yield loss and seed quality deterioration in the southern U.S. states of Arkansas, Louisiana, and Mississippi [11-13], although soybean cyst nematode is the major cause of crop damage in most parts of the country.”

We have also added a reference to a recent comprehensive review on the issue of the world-wide distribution of M phaseolina (Reference [14]) as follows: 

  1. Lodha S., Mawar R. Population dynamics of Macrophomina phaseolina in relation to disease management: A review. Journal of Phytopathology. 2019;00:1–17. DOI: 10.1111/jph.12854.

The reference numbers have been revised throughout the text to accommodate this addition. 

We feel it would be inappropriate in the context of a Brief Report in the Journal of Fungi to dwell any further on the motives of Mississippi soybean growers in supporting research on this devastating fungus. 

  1. Reviewer #2 felt that the following sentence (added at p. 2, l. 70, original MS to address a demand to include discussion of endophytes) was inappropriate:

“It was not determined if they had infected plants from the soil reservoir or were endophytes.”

We agree whole-heartedly that it was inappropriate and are pleased to remove it from the present version.  What was inappropriate was for a reviewer to demand that the manuscript address this tangential issue.  A Brief Report should be focused on a scientific discovery and not succumb to Reviewer demands to de-focus the narrative.  We feel that what was inappropriate was Reviewer 2’s demand that the manuscript be altered to address this tangential issue.  We have, in fact, carried out studies searching for M. phaseolina endophytes in three soybean cultivars grown in the region and failed to find any, although more than a dozen other fungal species were identified by DNA sequencing in the study.  Given that M. phaseolina has been shown to persist in soil for up to 15 years as a saprophyte (Kaur et al., 2012), the soil reservoir may be a more promising source for the isolates in this study.  These studies are not yet ready for publication, but if they were, we feel it would be inappropriate to de-focus this Brief Report by adding them to it. 

  1. Reviewer #2 felt that changes made in response to the request “Please underline more the aim of the studies” were not adequate. In response to this we have added an additional sentence to the description to the aim of the studies.  The current version of the manuscript on p. 2 at ll. 52-60 is as follows: 

It is not understood what factors specific to the region are responsible for the relatively greater pathogenicity of M. phaseolina there [14].

One plausible explanation for the greater frequency of charcoal rot disease in soybeans in this region is that M. phaseolina isolates endemic to the region produce unusual levels or diversity of mycotoxins. Therefore, as an approach to exploring the possible role of mycotoxin production in increased pathogenicity of M. phaseolina, the present study was undertaken to investigate the diversity of known mycotoxins and other secondary metabolites produced in culture by isolates from regional soybean plants symptomatic for charcoal rot using liquid chromatography tandem mass spectrometry (LC-MS/MS) with culture medium filtrates.  

We hope this addition makes the aims of the study clear.

  1. Reviewer #2 felt that describing occurrence in “>19% of cultures” as being “at relatively high frequency” needed explanation. At 19.1% of cultures, the metabolite was present at 3.82 times higher frequency than nine other metabolites.  We have removed the phrase being “at relatively high frequency” and now consider the issue to be mute.  We feel any further discussion of whether it is high or just higher would be a poor use of precious space in the Journal of Fungi

  1. Reviewer #2 requested the addition of a section to the Materials and Methods describing the statistical analysis of the frequency of occurrence of mycotoxins and other secondary metabolites in the study in more detail than was given in table footnotes. This we are pleased to do by adding the following section at p. 4, ll. 129-132:

2.5. Statitical Analysis

The frequencies of occurrence of mycotoxins and other secondary metabolites in the study were analyzed for significant differences by pair-wise applications of Fisher’s exact test to rank-ordered frequencies.  P<0.05 was considered significant.

  1. Reviewer #2 pointed out that the Tables were numbered out of order. We are pleased to correct this error throughout the manuscript. 

Round 3

Reviewer 2 Report

Dear Authors,

I have an honour to make a revision of the manuscript Brief communication entiled: “ First Report of the Production of Mycotoxins and Other Secondary Metabolites by Macrophomina phaseolina (Tassi) Goid. Isolates from Soybeans (Glycine max L.) Symptomatic with Charcoal Rot Disease” submitted to Journal of Fungi MDPI. But I am sorry to say that in “improved version of manuscript” again in the most points ignored my questions and concerns about the manuscript. Moreover, response for my comments are not written in cultural way and Authors name that all my comments “tangential issue”. I would like outline that if authors did not know if they investigate pathogen or not. Therefore, they are no reason to investigated it as pathogenesis or role of myttoxins . Ignoring reviewer comments to say in many points the Authors Response for reviewer questions is not deleting sentence in any means. Again is needed to make appropriate response for all my comments presented in second round of review.

Sincerely